# Collection of Bacterial Community Associated with Size Fractionated Aerosols from Kuwait

Nazima Habibi, Saif Uddin *, Fadila Al Salameen, Montaha Behbehani, Faiz Shirshikhar, Nasreem Abdul Razzack, Anisha Shajan and Farhana Zakir Hussain

Environment and Life Science Research Centre, Kuwait Institute for Scientific Research, Safat 13109, Kuwait; nhabibi@kisr.edu.kw (N.H.); fsalameen@kisr.edu.kw (F.A.S.); mbahbaha@kisr.edu.kw (M.B.); fshirshikar@kisr.edu.kw (F.S.); nabdulr@kisr.edu.kw (N.A.R.); ashajan@kisr.edu.kw (A.S.); fhussain@kisr.edu.kw (F.Z.H.)
* Correspondence: sdin@kisr.edu.kw

**Abstract:** Airborne particles play a significant role in the spread of bacterial communities. The prevalence of both pathogenic and non-pathogenic forms in the inhalable fractions of aerosols is known. The abundance of microorganisms in the aerosols heightens the likely health hazards due to inhalation since they serve as carriers for pathogens and allergens, often acting as a vector for pulmonary/respiratory infections. Not much information is available on the occurrence and prevalence of bacterial communities in different size-fractionated aerosols in Kuwait. A high-volume air sampler with a six-stage cascade impactor was deployed for sample collection at two sites representing a remote and an urban site. A total volume of $815 \pm 5$ m³ of air was passed through the filters to trap the particulate matter ranging from 0.39 to >10.2 μm in size (Stage 1 to Stage 5 and base filter). *Aeromonas* dominated all the stages at the urban site and Stage 5 at the remote site, whereas *Sphingobium* was prevalent at Stages, 2, 3 and 4 at the remote site. *Brevundimonas* were found at Stages 1 and 5, and the base filter at the remote site. These results show that the bacterial community is altered in different size fractions of aerosols. Stages 1–4 form the respirable fraction, whereas Stage 5 and particles on the base filter are the inhalable fractions. Many species of *Aeromonas* cause disease, and hence their presence in inhalable fractions is a health concern, meaning that species-level identification is warranted.

**Dataset:** The raw sequences obtained in this study are deposited in the NCBI database (bacterial sequences SRA: SUB6214874; BioProject: PRJNA766336 (accessions SRR16072385-SRR16072392).

**Keywords:** air metagenome; bioaerosols; QIIME; bacterial community

## 1. Summary

Aerosols are often loaded with biological materials such as fungal spores, bacteria, viruses, and plant and animal fragments [1,2]. The present dataset targets the presence of bacterial communities in aerosols. Information was generated as a result of high-throughput sequencing of aerosols collected over the course of one year from two sites representing different environmental milieu, i.e., a site in Kuwait City was designated as an urban site, while another site at the Kuwait–Iraq border on Abdally farm is considered as a remote site. The aerosol samples were collected using a high-volume air sampler (HVAS) equipped with a six-stage cascade impactor with aerodynamic sizes corresponding to 0.39 to 0.69 μm (base filter), >0.69 to 1.3 μm (Stage 5), >1.3 to 2.1 μm (Stage 4), >2.1 to 4.2 μm (Stage 3), >4.2 to 10.2 μm (Stage 2) and >10.2 μm (Stage 1). The samples were analyzed for microbial types and relative abundances (RA) through 16 s metagenomic sequencing. Raw sequences were filtered and processed through the Quantitative Insights into Microbial

Ecology (QIIME) software, yielding 12,797 operational taxonomic units (OTUs). Taxonomic profiling of these OTUs revealed 109 known bacterial genera to be distributed among the six size fractions. The RA of the genera varied in different size fractions. All the bacterial sequences were deposited on the public repository of the National Centre for Biotechnology Information (NCBI). These data provide insights into the size-fractionated bio-aerosols from the arid region. They further add value to assessing the microbial population associated with particles of 2.5 μm and smaller, as they are easily transported through the respiratory tract into the bronchioles and alveoli of the lungs and therefore pose a greater risk to human health.

## 2. Data Description

### 2.1. Study Area

The study area forms part of the hyper-arid region geographically covering Kuwait, which is located at the northern-eastern edge of the Arabian Peninsula, bordering Iraq to the north and Saudi Arabia to the south. Kuwait forms part of the regional wind corridor resulting in the transboundary transport of the microbial population into and via Kuwait. Because of its hyper aridity, low vegetation cover, and scarce precipitation, sandstorms are frequent. Considering the dominant northwest–southeast wind direction in Kuwait, two sampling sites covering different environmental milieu were selected for this study. A site in Abdally (30.05 N 47.71 E; 21 m above sea level) was referred to as a remote site, where the air mass from the north enters into Kuwait, and the other in Kuwait City (29.34 N 47.91 E; 1 m above sea level) where most of the urban population is concentrated, was referred to as an urban site (Figure 1). The remote location was on a private farm and the urban location was within the premises of our research institute.

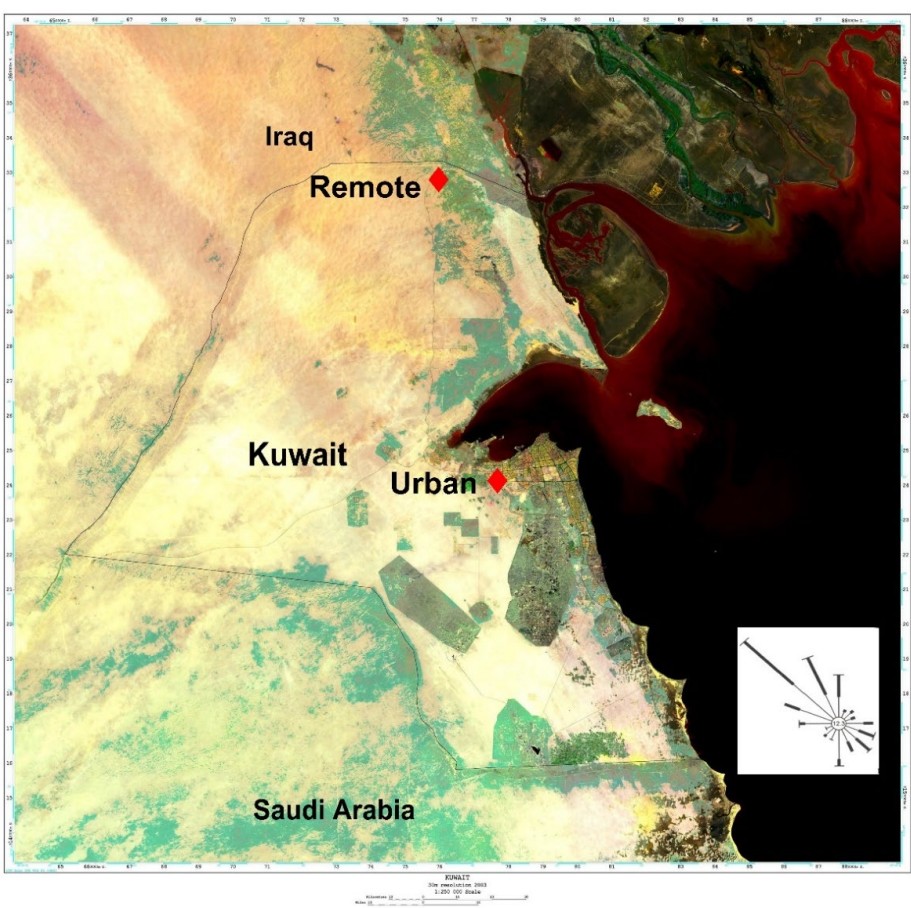

**Figure 1.** Location for sample collection using high-volume air samplers.

## 2.2. Bacterial Composition

On average (*n* = 34), 87% of the raw reads had a Phred score above Q > 20. Pre-processed reads from all the samples were pooled and clustered into OTUs based on their sequence similarity using Uclust. A total of 370,393 OTUs were identified from 6,755,671 reads. From a total of 370,393 OTUs, 341,215 OTUs with less than five reads were removed, and 29,178 OTUs were selected for taxonomic profiling. The OTUs with an inter-quartile range below 10%, the minimum count below 4, and sample prevalence below 60% were removed. This yielded 109 classified OTUs at the genus level from a total of 34 samples (Figure 2). The relative abundance (%) was calculated using each sample's OTU count number. *Aeromonas* was abundant in all the size fractions at the urban site. At the remote site, it was dominant only at Stage 5. *Brevundimonas* was abundant at Stages 1 and 7 at the remote site, whereas *Sphingobium* showed the maximum RA at Stages 2, 3 and 4. Only the top 10 genera were plotted on the graph, and the remaining were grouped as others. A significant proportion of these genera were recorded at all size fractions at both sites.

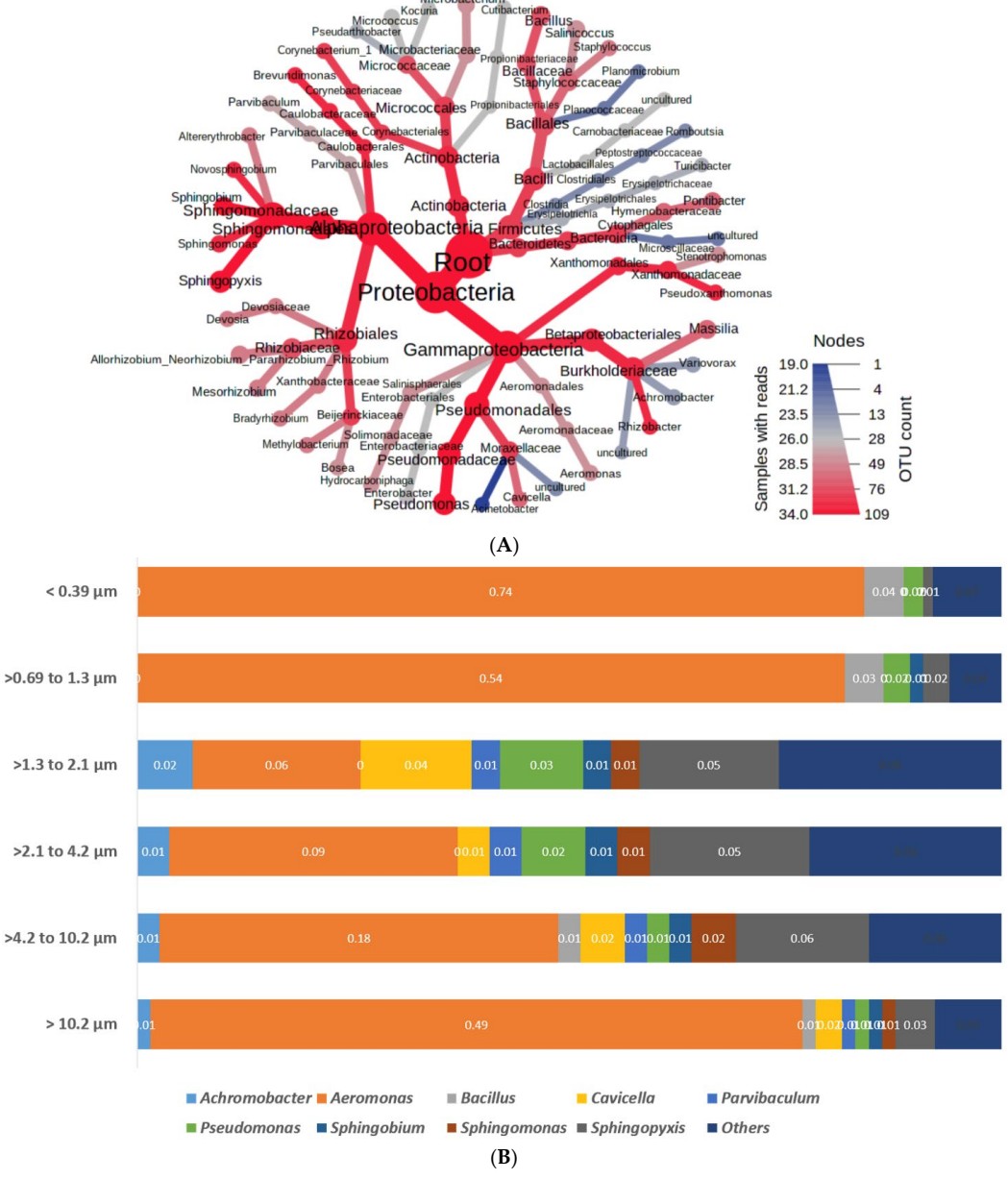

**Figure 2.** *Cont.*

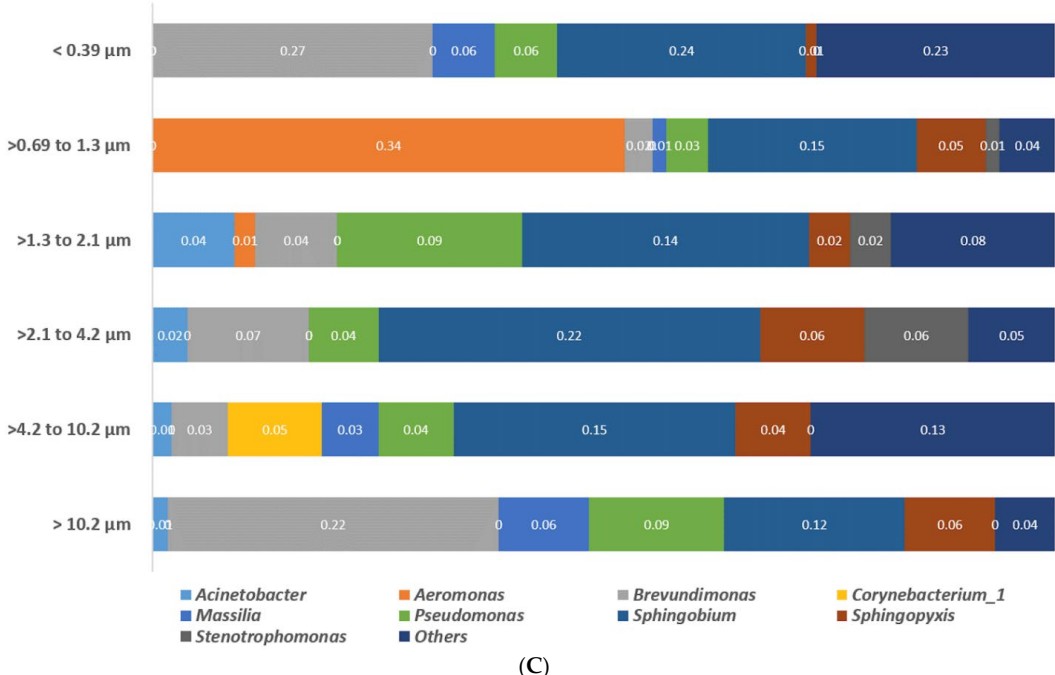

**Figure 2.** (**A**) Bacterial community composition of aerosols; (**B**) bacterial genera in different size fractions at the urban site; (**C**) bacterial genera in different size fractions at the remote site. The relative abundance of bacterial genera is plotted on the x-axis and the corresponding size fractions are presented on the y-axis. The top ten bacterial genera are plotted on the bar graphs and remaining grouped as others.

## 3. Methods

### 3.1. Sample Collection

The air samples were collected using an HVAS (Tisch Environmental, Inc., Cleves, OH, USA) equipped with a six-stage cascade impactor to fractionate the suspended PM into six distinct aerodynamic size classes (<0.39 to >10.2 μm). A known quantity of air was drawn using a pump and mass flow controller through a six-stage cascade impactor that had a clean sterilized slotted quartz filter (5.5″ × 5.5″) in each size fraction and a glass fiber filter (8″ × 10″) base filter at the end to trap particulate matter 0.69–0.39 μm in size [3–5]. The exact air volume for each sample was determined with the use of a calibrated magnehelic gauge (Tisch Environmental, Inc) to measure the pressure at the start and end of each sampling period [6] and a mass flow controller. A cascade impactor was deployed for 24 h at both sites. The sampling was performed for five months in November, December, February, July and September (2017–2018) to ascertain the background concentration of PM and the associated microbes from different trajectories. Calibration of the instruments was performed as per the manufacturer's guidelines, upon installation of the sampler, once every four months, and each time the motor was serviced [7]. The aerosol mass in each size class is given in Table 1. Along with the HVAS with cascade impactor, the particle mass counter was co-deployed at each site. The instruments were programmed to log data at 1 h intervals for the duration of sampling.

**Table 1.** Aerosol mass collected in 24-h time integrated sample.

| Date | Aerosol Mass in Grams | | | | | | |
|---|---|---|---|---|---|---|---|
| | Stage 1 | Stage 2 | Stage 3 | Stage 4 | Stage 5 | Stage 7 | Total |
| 12.11.2017 | 0.0095 | 0.0055 | 0.0209 | 0.0117 | 0.0098 | 0.1627 | 0.2201 |
| 14.11.2017 | 0.0103 | 0.0115 | 0.0223 | 0.0265 | 0.0188 | 0.4882 | 0.5776 |
| 12.12.2017 | 0.0112 | 0.0046 | 0.0073 | 0.0104 | 0.0211 | 0.3307 | 0.3853 |
| 05.02.2018 | 0.0150 | 0.0060 | 0.0168 | 0.0078 | 0.0105 | 0.3987 | 0.4548 |
| 03.07.2018 | 0.0020 | 0.0103 | 0.0215 | 0.0213 | 0.0114 | 0.3205 | 0.3870 |
| 09.09.2018 | 0.0047 | 0.0105 | 0.0060 | 0.0172 | 0.0086 | 0.2013 | 0.2483 |

### 3.2. DNA Extraction and Metagenomic Sequencing

DNA was isolated from all six stages, representing both the inhalable (stage 7 and Stage 5) and respirable (Stages 1–4) size fractions [1,8], by employing the Promega kit (Wizard Genomic, Madison, WI, USA). DNA purity (Absorbance ratio A260/A280) and quantity (Absorbance at 260 nm) were measured by the Nanodrop (Thermo Scientific, Carlsbad, CA, USA). The size and DNA integrity were assessed by means of agarose gel electrophoresis. All the DNA samples were also quantified through a Qubit fluorometer (Invitrogen, Carlsbad, CA, USA) The DNA concentration and yield is given in Table 2. The presence of bacterial DNA was checked through polymerase chain reaction employing the universal 16 s primers [2,9–11]. High-quality and -purity DNA was used for library preparation using the NEBNext Ultra DNA library preparation kit (New England BioLabs, France). The library quantification and quality estimation were done in Agilent 2200 TapeStation (Santa Clara, CA, USA). The prepared library was sequenced in Illumina HiSeq 2500 (San Diego, CA, USA) with 2 * 250 cycle chemistry.

**Table 2.** DNA concentration and yield in the bracket from Stages 1–5 and 7 of each sample.

| Date | DNA in ng $\mu L^{-1}$ | | | | | |
|---|---|---|---|---|---|---|
| | Stage 1 | Stage 2 | Stage 3 | Stage 4 | Stage 5 | Stage 7 |
| 12.11.2017 | 0.46 (9.24) | 1.13 (22.6) | 1.07 (21.4) | 0.46 (9.28) | 8.00 (160) | 2.20 (110) |
| 14.11.2017 | 0.72 (14.38) | 0.64 (12.82) | 0.60 (12.1) | 1.15 (23) | 0.13 (2.66) | 0.43 (8.52) |
| 12.12.2017 | 0.0 (0) | 0.29 (6) | 0.44 (8.82) | 0.74 (14.72) | 0.14 (2.7) | 1.92 (96) |
| 05.02.2018 | 6.27 (125.4) | 5.91 (118.2) | 8.47 (169.4) | 0.70 (13.9) | 1.90 (38.4) | 3.37 (168.5) |
| 03.07.2018 | 0.19 (3.9) | 0.68 (13.58) | 1.76 (8.56) | 1.13 (35.2) | 0.33 (6.54) | 0.74 (14.88) |
| 09.09.2018 | 11.5 (230) | 6.08 (121.6) | 1.17 (23.4) | 2.60 (52) | 2.07 (41.4) | 5.90 (118) |

*Total DNA yield is in nanograms.*

### 3.3. Bioinformatic Analysis

The raw reads obtained from the Illumina sequencing platform were demultiplexed and subjected to FastQC version 0.119 [12]. The base quality (Phred Score Q), base composition, GC content, ambiguous bases (other than A,T,G,C), and adapter dimers were thoroughly checked prior to the bioinformatics analysis. The forward V3 specific primer and reverse V4 specific primers were trimmed using PERL scripts. The properly paired-end reads with Phred score quality (Q) > 20 were considered for V3-V4 consensus generation. The reads were merged using FLASH program (version 1.2.11) with a minimum overlap of 10 bp to a maximum overlap of 240 bp with zero mismatches [13]. All reads with an average contig length of 350 to 450 bp (V3-V4 region) were used to make a consensus sequence. The selection of operational taxonomic units (OTU) and taxonomy classification was performed using the pre-processed consensus sequences. Pre-processed reads from all

samples were pooled and clustered into OTUs based on their sequence similarity using the Uclust [14] program (similarity cutoff = 0.97) available in QIIME software [15].

**Supplementary Materials:** The following are available online at https://figshare.com/s/75fcc36c9 0e524b5628f accessed on 9 November 2021.

**Author Contributions:** Conceptualization, S.U.; Methodology, N.H., N.A.R., A.S., F.Z.H.; software, N.H.; validation, N.H. and S.U.; formal analysis, S.U.; investigation, F.S., F.A.S.; resources, S.U., F.A.S., M.B.; writing—original draft preparation, N.H.; writing—review and editing, S.U.; visualization, N.H.; supervision, S.U.; project administration, F.A.S., M.B.; funding acquisition, S.U., N.H., M.B. All authors have read and agreed to the published version of the manuscript.

**Funding:** This research was funded by grants from the Kuwait Foundation for Advancement of Research (KFAS) PN20-43BO-01 and PR18-14SE-01.

**Institutional Review Board Statement:** Not applicable.

**Informed Consent Statement:** Not applicable.

**Data Availability Statement:** The data presented in this study are available within the Supplementary Materials at https://figshare.com/s/75fcc36c90e524b5628f accessed on 9 November 2021. Raw sequences are deposited at the NCBI public repository https://dataview.ncbi.nlm.nih.gov/object/ PRJNA766336?reviewer=ana1j8l4j54t12ng6nifgg0ck, accessed on 9 November 2021.

**Acknowledgments:** We thank Hasan Al-Shemmari, Kuwait Institute for Scientific Research, Kuwait for providing space on his farm to install the high-volume air sampler (HVAS) and supplying the electricity for its operation throughout the study.

**Conflicts of Interest:** The authors declare no conflict of interest.

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
