# Peer review of "Collection of Bacterial Community Associated with Size Fractionated Aerosols from Kuwait"

_data, 2021_

Round 1

Reviewer 1 Report

The technical note entitled "Collection of Bacterial Community Associated with Size Fractionated Aerosols from Kuwait" is a very concise description of the data obtained for two different locations. 

The methodology is classical and robust, this is , in my opinion, the sign that the data acquired are reliable. 

i would only advise the authors to also give a granulometric distribution of the whole aerosol acquired (by weighing the sampled aerosol for each stage). This would help the reader to link the data presented with the overall aerosol fraction in those sites. 

A second recommandation would concern the comparison between the biological fraction of the aerosol and the "non-biological" fraction. As it is mentionned in the introduction, the fact that particles could carry micro-organismes remains an opened question. 

Author Response

Reviewer 1

Comment

Reply

The technical note entitled "Collection of Bacterial Community Associated with Size Fractionated Aerosols from Kuwait" is a very concise description of the data obtained for two different locations.

We thank the reviewer for support of this study.

The methodology is classical and robust, this is, in my opinion, the sign that the data acquired are reliable. 

Appreciate the support, we believe we have adopted a standard methodology and produced reliable quality-assured data.

I would only advise the authors to also give a granulometric distribution of the whole aerosol acquired (by weighing the sampled aerosol for each stage). This would help the reader to link the data presented with the overall aerosol fraction in those sites. 

Thanks for the suggestion, we have included Table 1 that has the mass of each aerosol fraction, and Tabel 2 that provides information on the amount of DNA retrieved from each size fraction and the yield.

A second recommendation would concern the comparison between the biological fraction of the aerosol and the "non-biological" fraction. As it is mentioned in the introduction, the fact that particles could carry micro-organisms remains an opened question. 

We have actually submitted another article on this aspect looking at bioaerosol. Since we are privy to the information about 5 – 10% of the total mass is microbial load. The rest is a non-biological fraction basically the inorganic phase.

Reviewer 2 Report

The authors describe the collection of airborne bacteria for two sites in Kuwait over the period of a year followed by metagenomic analysis to ascertain the distribution of bacterial species in the samples. To a non-expert the manuscript appears to describe the data collection and analysis well.

One specific point was a lack of information about the times of the sampling and how the collected fractions were dealt with. It would be good to know if the fractions collected at different times were pooled and analysed together or if the genetic analysis was done on each sample and time point separately and what, if any, averaging was done.

Also, the text in the figures is almost impossible to read and should be made larger.

Author Response

Reviewer 2

Comments

Reply

The authors describe the collection of airborne bacteria for two sites in Kuwait over the period of a year followed by metagenomic analysis to ascertain the distribution of bacterial species in the samples. To a non-expert the manuscript appears to describe the data collection and analysis well.

The manuscript provides information on bacterial diversity in different size fractions of the aerosol.

One specific point was a lack of information about the times of the sampling and how the collected fractions were dealt with. It would be good to know if the fractions collected at different times were pooled and analysed together or if the genetic analysis was done on each sample and time point separately and what, if any, averaging was done.

The samples were collected using an HVAS sampler equipped with a six-stage cascade impactor. The information is provided in the manuscript. However, a new Table 1 is also included that provides information on the same collection date.

Also, the text in the figures is almost impossible to read and should be made larger.

Higher-resolution figures are provided.